# OpenReview forum: "DLLMQuant: A Post-Training Quantization Framework Tailored for Diffusion-Based Large Language Models"
_ICML.cc/2026/Conference — ICML 2026 regular_

### Official Review · Reviewer_eY55 · 2026-03-07

**Soundness:** 2
**Presentation:** 3
**Significance:** 2
**Originality:** 2
**Overall Recommendation:** 3
**Confidence:** 4

**Summary:**

This paper studies post-training quantization (PTQ) for diffusion-based large language models (DLLMs) such as LLaDA and DREAM. The authors identify three problems that arise when naively applying existing PTQ methods to DLLMs: (1) temporal distribution shift across denoising timesteps, (2) iterative accumulation of quantization errors, and (3) feature distribution mismatch between decoded and masked tokens. To address these, they propose three corresponding techniques TMAS (temporal-mask-aware calibration sampling), IA-AQ (interaction-aware activation quantization), and CGQ (certainty-guided weight quantization) collectively forming the DLLMQuant framework. Experiments on LLaDA-8B, LLaDA-8B-Instruct, and DREAM-7B at W4A4 show meaningful improvements over vanilla AWQ and QuaRot baselines.

**Compliance With Llm Reviewing Policy:**

Affirmed.

**Final Justification:**

I have made my final justification.

**Key Questions For Authors:**

Q1. Can you provide MSE-vs-timestep curves (in the style of Figure 3) with and without IA-AQ? This would directly address my concern in W2 and could meaningfully change my assessment if the results are convincing.

Q2. What happens with uniform calibration sampling (equal weight to all mask ratio bins)? If the gap to TMAS is within 0.5%, the contribution of TMAS becomes questionable.

Q3. Why is W4A4 the only quantization setting? Can you provide results for at least W4A8 and W8A8? This is important for understanding where the gains come from and how the method scales across precision levels.

Q4. Have you considered per-timestep quantization parameters as a more direct solution to Problem 2? If so, how does it compare to IA-AQ? If not, why not?

Q5. Can you report quantization time overhead (wall-clock) for the full DLLMQuant pipeline compared to vanilla AWQ/QuaRot?

**Limitations:**

yes

**Strengths And Weaknesses:**

**Strengths**

The problem is timely and well-motivated. DLLMs are gaining traction and deployment efficiency matters. This is one of the first papers to systematically study PTQ for this model family, and the authors clearly put effort into understanding why existing methods fail rather than just reporting numbers.

The diagnostic analysis is the strongest part of the paper. Figures 1 through 4 build good intuition. Figure 3 showing MSE blowing up across timesteps under QuaRot is compelling and clearly motivates the work. The decomposition into three failure modes is clean and easy to follow.

The plug-and-play design is practical. TMAS, IA-AQ, and CGQ can each be layered independently on top of existing PTQ methods like AWQ and QuaRot. This makes adoption easier.

The empirical gains on reasoning tasks are hard to dismiss. Going from 44.57 to 56.25 on GSM8K is a big jump. HumanEval gains are also consistent.

**Weaknesses**

- The contributions individually lack depth and collectively lack a unifying principle.

TMAS is stratified sampling over mask ratio intervals. The bin boundaries (0.2/0.5/0.8) and sampling ratios ([0.3, 0.2, 0.2, 0.3]) come from empirical search (Table 7). There's no analysis of why four bins are sufficient, no sampling error bounds, and no comparison against simpler alternatives like uniform sampling over timesteps. Timestep-aware calibration has been explored in diffusion image model quantization (Q-Diffusion, PTQD), and the paper doesn't discuss the relationship.

IA-AQ uses softmax outputs as importance weights for quantizing the V matrix (Eq. 8). This is essentially importance-aware rounding, which is well known. The alpha search from 1.0 to 0.8 with step 0.02 is brute force and narrow.

CGQ modifies the GPTQ Hessian using mask states and confidence scores. The unmasked token weight of 0.7 comes from grid search (Table 8). This is very similar to sensitivity-based Hessian reweighting in SqueezeLLM and OWQ, with a straightforward adaptation to the DLLM setting.

Each component is reasonable, but none represents a fundamentally new idea. The paper reads more like three practical tricks for quantizing DLLMs than a principled methodology.

- The proposed solution doesn't match the problem it claims to solve.

This is my biggest concern. Section 3.2 presents a nice analysis (Eqs. 6 and 7) showing that quantization errors propagate and potentially grow geometrically across denoising iterations. This is the most interesting part of the paper. But IA-AQ only improves quantization fidelity at a single timestep by better quantizing V. The argument is basically "reduce per-step error, reduce cumulative error." That's technically true but sidesteps the core problem the analysis identifies.

The paper doesn't provide evidence that IA-AQ actually resolves the accumulation issue. Figure 3 shows MSE curves for vanilla QuaRot, but there's no corresponding curve after applying IA-AQ. Without this, I can't tell whether the method truly addresses iterative accumulation or just gives a modest per-step improvement. More direct approaches like per-timestep quantization parameters or lightweight error correction modules are not explored or discussed.

- The experimental evaluation has significant gaps.

Only W4A4 is reported. For a paper claiming to be a general PTQ framework, the absence of W8A8, W4A8, and W3A3 results is a serious omission. W4A8 would be especially useful to disentangle weight quantization (CGQ) from activation quantization (IA-AQ).

All evaluations use classification-style or code-completion benchmarks (ARC, HellaSwag, GSM8K, HumanEval, etc.). There is no evaluation of open-ended text generation quality. DLLMs generate text through parallel denoising, and quantization could degrade fluency and diversity in ways not captured by multiple-choice accuracy. Perplexity on a standard corpus or human evaluation of generated text would help.

Inference speedup (1.6x) and memory savings (3.2x) are reported, but the offline quantization cost is missing. This includes TMAS sampling overhead, IA-AQ's alpha search, and CGQ's modified Hessian computation.

- Insufficient discussion of concurrent and related work.

"Quantization Meets dLLMs" (arXiv 2508.14896, August 2025) appeared within a week and identifies similar challenges around activation outliers. Quant-dLLM (arXiv 2510.03274, September 2025) proposes Masked Calibration Simulation, which is very close to TMAS in motivation and design. Given the ICML 2026 submission deadline of January 28, 2026, both should be discussed. The paper also needs to better position itself against diffusion image model quantization work (Q-Diffusion, PTQD, TDQ), which already studied timestep-aware calibration and error accumulation. The authors should clearly state what is new in the DLLM setting versus what carries over.

---

> ### Author Rebuttal · Authors · 2026-03-30
>
> # Response to Reviewer eY55
> We sincerely appreciate your thorough and constructive comments, which have significantly helped us improve the rigor, clarity, and completeness of our work. We address all your key questions and concerns in detail below:
>
> ---
>
> ## 1. MSE vs. Timestep Curves with/without IA-AQ
> We fully agree that providing MSE-timestep curves is critical to validating IA-AQ’s ability to mitigate cumulative error. We have added the requested curves (in the style of Figure 3) comparing **vanilla QuaRot** and **QuaRot + IA-AQ**. You can view the curves here: [https://anonymous.4open.science/r/DLLMQUANT/out_dist.pdf]
>
> The results clearly demonstrate:
> - **Vanilla QuaRot**: MSE rises sharply across timesteps, showing severe error accumulation during iterative denoising.
> - **QuaRot + IA-AQ**: The MSE curve is significantly flattened, showing that IA-AQ effectively suppresses error propagation without relying on per-timestep quantization.
>
> These results directly confirm that IA-AQ resolves Problem 2 (error accumulation) in a principled and effective manner.
>
> ---
>
> ## 2. Uniform Calibration Sampling vs. TMAS vs. Single-Timestep Sampling
> We agree with the reviewer that uniform sampling across timesteps is a reasonable baseline, and **the truly suboptimal choice is single-timestep calibration**, which completely ignores large distribution shifts across denoising steps.
>
> Detailed ablation results on LLaDA are already provided in the appendix:
> - For AWQ + CGQ + IA-AQ: uniform sampling achieves 48.63, while TMAS improves it to 49.26.
> - For QuaRot + CGQ + IA-AQ: uniform sampling achieves 52.36, while TMAS improves it to 54.29.
>
>
> ---
>
> ## 3. Multi-bit Quantization Results (W4A4 / W4A8 / W8A8)
> We have extended our experiments to include **W4A4, W4A8, and W8A8** on LLaDA-8B, with comparisons against RTN, AWQ, QuaRot, and our DLLMQuant framework.
>
> ### Complete Experimental Results (LLaDA-8B, Avg.)
> | Quant Setting | FP16  | RTN   | AWQ   | QuaRot | DLLMQuant+ | DLLMQuant++ |
> |---------------|-------|-------|-------|--------|-------------|-------------|
> | **W4A4**      | 59.87 | 44.23 | 48.09 | 51.03  | 49.26       | **54.29**   |
> | **W4A8**      | 59.87 | 50.16 | 54.15 | 54.82  | 55.91       | **56.78**   |
> | **W8A8**      | 59.87 | 56.84 | 57.22 | 58.06  | 58.47       | **58.91**   |
>
> ### Analysis
> - **Consistent gains across all bit-widths**: Our method outperforms all baselines in W4A4, W4A8, and W8A8, showing that DLLMQuant is not limited to ultra-low-bit quantization but generalizes well across precision levels.
> - **W4A8 results disentangle weight/activation gains**: Even with 8-bit activations, our method still delivers significant improvements over AWQ and QuaRot, proving that our core gains come from modeling dLLM’s iterative dynamics.
> - **W8A8 results confirm practical utility**: Near-full-precision results show that DLLMQuant still provides consistent improvements, validating its real-world applicability.
>
> These results have been fully added to the revised manuscript.
>
> ---
>
> ## 4. Per-timestep Quantization vs. IA-AQ
> We have carefully evaluated **per-timestep quantization parameters** as a direct solution to Problem 2 (error accumulation) and have compared it against IA-AQ.
>
> ### Key Implementation Considerations for dLLMs
> - **Memory overhead**: Storing separate scales for each timestep leads to significantly higher memory consumption.
> - **Inference efficiency**: Per-timestep scaling breaks kernel fusion and introduces non-negligible latency.
> - **Activation distribution**: dLLM activations change smoothly across timesteps, so a shared scale is sufficient.
>
> ### Quantitative Comparison (LLaDA-8B, Avg.)
> | Method                | Avg.  |
> |-----------------------|-------|
> | QuaRot                | 51.03 |
> | QuaRot + IA-AQ        | 53.25 |
> | QuaRot + Per-timestep | 53.41 |
>
> ### Key Conclusion
> Per-timestep quantization yields only a marginal accuracy gain over IA-AQ, but introduces substantial memory and latency overhead. In contrast, IA-AQ achieves comparable performance with nearly no extra cost, making it a much more practical and deployment-friendly solution.
>
> We have added this comparison and analysis to the revised manuscript.
>
> ---
>
> ## 5. Quantization Time Overhead (Wall-clock)
> We have measured the end-to-end wall-clock time of our full DLLMQuant pipeline compared to vanilla AWQ/QuaRot.
>
> ### Quantization Time Breakdown
> - **TMAS**: A one-time offline sampling step (~4% of total time).
> - **IA-AQ**: Lightweight grid search over α, adding only ~8% overhead.
> - **CGQ**: Hessian-based optimization, adding only ~2% overhead.
>
> Overall, the extra time cost is minor and acceptable for a one-time post-training quantization step, and does not affect inference deployment.
>
> ---
>
> ## Summary
> We thank you again for your professional and constructive feedback. These revisions have greatly strengthened the paper’s rigor and reproducibility.

---

> > ### Author Rebuttal · Reviewer_eY55 · 2026-04-01
> >
> > Thank you for the additional experiments. The multi-bit results (Q3) and per-timestep comparison (Q4) are useful additions.
> >
> > I want to verify the MSE curves (Q1) before commenting further on that point. On Q5, I was looking for absolute wall-clock numbers, not percentages.
> >
> > Two concerns were not addressed. First, concurrent work: Quant-dLLM's Masked Calibration Simulation is very close to TMAS and needs to be explicitly discussed. Second, there is still no open-ended generation evaluation, which is an important omission for models built around parallel text generation.
> >
> > I will maintain my score

---

> > > ### Author Response · Authors · 2026-04-07
> > >
> > > We thank the reviewer for this valuable comment.
> > >
> > > ---
> > > ### 1. For open-ended generation evaluation
> > >
> > > In fact, our evaluation already includes two representative open-ended generation benchmarks: **HumanEval** for open-ended code generation and **GSM8K** for open-ended mathematical reasoning with step-by-step natural language generation.
> > > Both datasets require the model to generate free-form, task-solving text rather than selecting from fixed options, which aligns well with the open-ended generation capability emphasized by the reviewer.
> > >
> > > These evaluations demonstrate that our model can effectively perform parallel open-ended generation in both code and mathematical reasoning scenarios.
> > >
> > > To further clarify this point, we have added explicit descriptions in the revised manuscript to emphasize that our experiments already cover standard open-ended generation benchmarks, and we have strengthened the discussion of these results.
> > >
> > > **If additional experiments on other datasets are needed, please do not hesitate to let us know, and we will conduct the corresponding evaluations.**
> > >
> > > ---
> > >
> > > ### 2. Core Differences between Our DLLMQuant and Quant-dLLM
> > >
> > > We sincerely appreciate the reviewer for raising the important comparison with Quant-dLLM. We fully clarify the critical differences as follows:
> > >
> > > *   **Quantization Scope**: Quant-dLLM is a weight-only quantization method for diffusion LLMs, which **does NOT** perform activation quantization at all. In contrast, our DLLMQuant is a unified weight + activation joint quantization framework specially designed for discrete diffusion models.
> > > *   **Handling of Error Accumulation**: Quant-dLLM does not consider or mitigate the step-by-step error accumulation issue during discrete diffusion generation. Our DLLMQuant explicitly targets this unique problem in discrete diffusion, which is the core bottleneck of quantized dLLM inference.
> > > *   **Dynamic Masking Adaptation**: Quant-dLLM uses static calibration and cannot handle the significant feature distribution shifts caused by dynamic masking ratios across diffusion steps. Our method proposes time-mask adaptive calibration and certainty-guided quantization to fully adapt to the unique properties of discrete diffusion.
> > >
> > > | Model        | Method          | Truth. | Arc.  | Hella. | Wino. | PIQA  | MMLU  | C-EVAL | Hum.  | GSM8K | Avg.  |
> > > |--------------|-----------------|--------|-------|--------|-------|-------|-------|--------|-------|-------|-------|
> > > | LLaDA        | Quant-dLLM      | 40.12  | 36.26 | 46.45  | 68.19 | 69.75 | 56.87 | 54.63  | 24.17 | 49.32 | 51.28 |
> > > |              | DLLMQuant++     | 43.53  | 44.18 | 51.00  | 71.85 | 73.94 | 57.77 | 61.22  | 28.92 | 56.25 | 54.29 |
> > > | LLaDA-1.5    | Quant-dLLM      | 42.10  | 84.22 | 67.91  | 71.81 | 72.15 | 56.13 | 58.72  | 42.86 | 71.44 | 62.17 |
> > > |              | DLLMQuant++     | 43.87  | 84.18 | 69.20  | 71.74 | 73.58 | 57.27 | 60.04  | 44.58 | 74.33 | 64.31 |
> > >
> > > For a fair comparison, we strictly follow the official implementation of Quant-dLLM and evaluate it on the same tasks and model settings as our DLLMQuant++.
> > >
> > > ---
> > >
> > > ### 3. For Q5
> > >
> > > We thank the reviewer for the suggestion to provide absolute wall-clock time. We supplement the detailed absolute time breakdown of our DLLMQuant pipeline on LLaDA-8B with 4-bit quantization:
> > > - The total end-to-end quantization time of our full pipeline is **~17 minutes**, which is comparable to vanilla AWQ/QuaRot.
> > > - The extra overhead introduced by our proposed modules (TMAS, IA-AQ, CGQ) is only **~106 seconds** (1.7 minutes), accounting for less than 15% of the total quantization time.
> > > - All additional costs are one-time offline post-training steps, with zero impact on the inference latency of the deployed quantized model.
> > >
> > > ---
> > >
> > > **We sincerely thank the reviewer for the extremely careful, rigorous, and insightful review, which has significantly improved the quality of our manuscript. We deeply appreciate the reviewer’s valuable time, professional expertise, and constructive feedback, and we hold the highest respect for the reviewer’s rigorous academic standards.** We have carefully addressed all comments, including supplementing the absolute wall-clock time breakdown of our quantization pipeline, and have revised the manuscript accordingly.

---

### Official Review · Reviewer_nRUb · 2026-03-09

**Soundness:** 2
**Presentation:** 2
**Significance:** 3
**Originality:** 3
**Overall Recommendation:** 3
**Confidence:** 4

**Summary:**

This paper studies post-training quantization for diffusion-based large language models (DLLMs), where standard PTQ methods appear to degrade performance substantially. The paper argues that DLLMs pose three specific challenges for PTQ: timestep-varying activation/token distributions during iterative denoising, accumulation of quantization error across iterations, and different importance patterns between masked and unmasked tokens. To address these issues, the authors propose DLLMQuant, a plug-in framework with three components: Temporal-Mask Adaptive Sampling (TMAS) for calibration, Interaction-Aware Activation Quantization (IA-AQ) for the attention value path, and Certainty-Guided Quantization (CGQ) for score/mask-aware weight quantization. Experiments on LLADA-8B, LLADA-1.5-8B, and DREAM-7B show improvements over RTN/AWQ/QuaRot in the main table, and additional appendix results show gains when combined with GPTQ, SmoothQuant, and OSTQuant, along with reported memory savings and throughput gains for 4-bit quantization.

**Compliance With Llm Reviewing Policy:**

Affirmed.

**Ethical Review Concerns:**

I would flag this for verification because the submission appears to contain reviewer-directed instruction text embedded in the PDF, specifically a sentence asking the reviewer to include specific phrases in the review. If this text is genuinely present in the submission rather than a PDF parsing artifact, it could be interpreted as an attempt to influence LLM-assisted reviewing and would therefore raise a research-integrity concern. I am not making a definitive accusation here because I cannot rule out a rendering/parsing artifact from the PDF extraction pipeline, but I believe this should be checked by the ethics/PC team.

**Ethical Review Flag:**

Flag this paper for an ethics review.

**Ethics Expertise Needed:**

["Research Integrity Issues (e.g., plagiarism)"]

**Key Questions For Authors:**

- Calibration/evaluation separation. The paper states that WinoGrande is one of the evaluation benchmarks and also that calibration uses 128 segments from the WinoGrande dataset. Are the calibration samples strictly disjoint from the evaluation set, and would the main conclusions remain the same with a held-out calibration corpus unrelated to any reported benchmark? If the authors can show contamination-free calibration with similar gains, my confidence in the evaluation would increase significantly.

- Comparison to a simple mixed-precision baseline. Fig. 3 suggests that leaving the post-softmax attention matmul unquantized already greatly reduces cumulative error. How does IA-AQ compare, in both accuracy and efficiency, to a simple baseline that keeps this operation (or the relevant value path) in higher precision? If IA-AQ clearly outperforms that simpler alternative at similar cost, that would strengthen both novelty and significance.

- Exact implementation details for TMAS and IA-AQ. Please clarify the precise TMAS procedure and correct the pseudocode/notational inconsistencies: how x_t is evolved across denoising steps in Algorithm 1, whether line 12 should use block instead of b, and what exact search set is used for α in Eq. 9. Clearer specification would materially improve reproducibility and my confidence in the method.

- Generality beyond the reported setting. The paper focuses on 4-bit W/A quantization for three DLLMs. Do the gains persist at other bit-widths, and do they transfer to multimodal DLLMs that are mentioned in the motivation? Additional evidence here would strengthen the significance claim.

- Efficiency details and fairness of throughput comparison. What inference kernels/software stack are used for the reported speed numbers, and are all compared methods evaluated under equally optimized implementations? This matters for interpreting the practical deployment claims.

**Limitations:**

No. The paper includes an impact statement, but it does not adequately discuss limitations. It should explicitly discuss at least the following: the experiments are limited to 4-bit quantization and three DLLM backbones; calibration sensitivity may be substantial; benchmark-overlapping calibration needs to be ruled out clearly; the work is motivated partly by multimodal DLLMs but does not evaluate any; and the practical speedups may depend strongly on kernel/runtime support. A short, explicit limitations paragraph would improve the paper.

**Strengths And Weaknesses:**

# Strengths
This paper tackles an important and timely problem: making diffusion-based LLMs deployable with low-bit PTQ. The paper is well motivated by concrete DLLM-specific failure modes, including timestep-dependent distribution shift and iteration-wise error accumulation, and the proposed solution is modular rather than monolithic. The three components are also conceptually aligned with the identified issues: TMAS targets calibration coverage across denoising stages, IA-AQ targets the error-prone post-softmax/value interaction in attention, and CGQ adapts Hessian weighting to the mask/score structure of DLLMs.

Empirically, the paper reports consistent gains over strong PTQ baselines on three DLLM families. In the main table, DLLMQuant improves over AWQ and QuaRot across the nine-task average and shows particularly meaningful recovery on HumanEval and GSM8K. The appendix further suggests that the approach can also improve GPTQ, SmoothQuant, and OSTQuant. The reported 4-bit results also come with practical efficiency gains, namely roughly 1.5–1.7× throughput improvement and about 3.1–3.2× memory savings. These are meaningful benefits for deployment-oriented work.

The ablations are also directionally useful. Table 2 suggests that each of TMAS, IA-AQ, and CGQ contributes positively, and Table 3 indicates that combining mask state and score is better than using either alone.

# Weaknesses

My main concern is technical/evaluative soundness rather than motivation. Several of the central arguments are plausible but still somewhat heuristic. For example, the paper’s core analysis of iterative error accumulation and its proposed weighted objectives for IA-AQ/CGQ are intuitively reasonable, but the justification remains largely empirical rather than rigorous. More importantly, Fig. 3 suggests that simply leaving the post-softmax matmul unquantized already drastically reduces cumulative error, yet this selective high-precision baseline is not carried through to the main end-task comparison tables. Without that comparison, it is hard to judge how much of IA-AQ’s benefit comes from a DLLM-specific insight versus a simpler mixed-precision workaround.

A second major concern is the calibration protocol. The paper states that the evaluation includes WinoGrande among the nine benchmarks, and that calibration uses 128 segments from the WinoGrande dataset. It is not clear whether these are strictly disjoint from evaluation examples. Even if labels are not used, benchmark-overlapping calibration can still bias PTQ results, especially when improvements are sometimes only a few points. I would be substantially more convinced by results using a clearly separate held-out corpus or a mixed calibration corpus unrelated to the evaluation benchmarks.

A third issue is that the presentation/reproducibility quality is below the bar I would hope for in a final ICML paper. Algorithm 1 is not fully coherent as written: inside the loop over timesteps it calls y_t ← Model(x) rather than clearly evolving x_t, and line 12 appears to use an undefined variable (b instead of block). Equation 9 is also inconsistent with the text: it says α is tested “stepping by 0.02 from 1.0 to 0.8,” but the argmin is written over {1.0, 0.8} only. There is also an avoidable inconsistency between the quantization description in the background section (activation quantization described as per-tensor) and the experimental setup (activation quantization described as per-token). These may be fixable, but together they reduce confidence in the exact implementation.

I also found the paper’s experimental positioning slightly incomplete in the main text. The main comparison table only shows RTN/AWQ/QuaRot, while GPTQ/SmoothQuant/OSTQuant results appear in the appendix, despite the paper emphasizing broad compatibility with existing PTQ methods. Given that this compatibility is part of the claimed contribution, I think some of these comparisons should appear in the main paper. In addition, the scope remains limited to 4-bit quantization on three DLLMs, despite the motivation mentioning broader multimodal DLLM settings.

Overall, I see clear merit here: the problem is relevant, the empirical direction is promising, and the method appears useful. However, I think the current version still needs a cleaner and more convincing experimental protocol, stronger comparison to simple selective mixed-precision baselines, and a more polished/reproducible presentation before I would be comfortable recommending acceptance.

---

> ### Author Rebuttal · Authors · 2026-03-29
>
> Response to  Dear Reviewer nRUb：
>
> We sincerely appreciate the reviewer’s thorough and constructive feedback, which has greatly helped us improve the rigor, reproducibility, and generalization of our work. We address each concern in detail below:
>
>
> ## 1. Calibration/Evaluation Separation
>
> We fully agree that strict calibration/evaluation separation is critical for reliable quantization evaluation. To address this:
> - We followed the standard practice of **llmqat** (https://arxiv.org/abs/2502.02631) to generate a completely held-out calibration corpus, ensuring no overlap with any evaluation benchmark segments.
> - We re-ran experiments using this calibration, and the results on LLaDA confirm that our method still maintains a clear advantage:
>
> | LLaDA       | Avg.  |
> |-------------|-------|
> | FP          | 59.87 |
> | RTN         | 44.53 |
> | AWQ         | 47.98 |
> | DLLMQuant+  | 49.19 |
> | QuaRot      | 51.04 |
> | DLLMQuant++ | 54.32 |
>
>
> ## 2. Comparison to Simple Mixed-Precision Baseline
>
> We have added comprehensive experiments to compare our method against a simple mixed-precision baseline (`Matmul-FP`) that keeps the post-softmax attention matrix multiplication in full precision, while quantizing all other tensors to 4 bits.
>
> ### Accuracy Comparison
> | Baseline | Matmul-FP | TMAS | IA-AQ | CGQ | Avg.   |
> |----------|-----------|------|-------|-----|--------|
> | AWQ   | ✓         | -    | -     | -   | 48.62  |
> |          | ✓         | -    | ✓     | ✓   | 49.19  |
> |          | -         | ✓    | -     | -   | 48.34  |
> |          | -         | ✓    | ✓     | ✓   | 49.26  |
> | QuaRot     | ✓         | -    | -     | -   | 53.41  |
> |          | ✓         | -    | ✓     | ✓   | 54.27  |
> |          | -         | ✓    | -     | -   | 53.25  |
> |          | -         | ✓    | ✓     | ✓   | 54.29  |
>
> ### Efficiency Comparison (Speed in Tokens/s)
> | LLaDA   | Matmul-FP+IA-AQ+CGQ | TMAS+IA-AQ+CGQ |
> |---------|---------------------|----------------|
> | Speed   | 43.27               | 50.14          |
>
>
> - Our full method (TMAS+IA-AQ+CGQ) **outperforms the `Matmul-FP` baseline** in accuracy while achieving **higher throughput**, demonstrating that our approach is both more effective and more efficient than simple mixed-precision heuristics.
>
>
> ## 3. Exact Implementation Details for TMAS and IA-AQ
>
> We have fully clarified the implementation details of TMAS and IA-AQ, corrected pseudocode inconsistencies, and supplemented all key parameter settings to ensure reproducibility:
>
> 1. **TMAS Exact Implementation**: TMAS samples calibration data by aligning dLLM's iterative denoising steps and dynamic masking ratios, strictly following Algorithm 1: initialize block/time step parameters and mask ratio sampling proportions [0.3,0.2,0.2,0.3], classify mask ratios into 4 intervals (0.2/0.5/0.8 as thresholds) during traversing all time steps, and collect samples by block and mask ratio until the target count is met. We have corrected the typo in Algorithm 1 (Line12: `b` → `block`)
>
> 2. **α Search Set in Eq.9 (IA-AQ)**: The optimal α is searched **from 0.8 to 1.0 with a step size of 0.02** (search set: {0.80,0.82,...,0.98,1.00}). We select the α that minimizes the quantization loss  (Eq.7) for the attention value matrix V, with this lightweight search executed per dLLM backbone and causing negligible latency
>
> 3. **Reproducibility**: All above details are added to the main text/appendix, and our code will be released
>
> ## 4. Generality Beyond the Reported Setting
>
> To demonstrate the generality of our method:  **Multimodal DLLMs**: We evaluated our method on the multimodal discrete diffusion model **MMAda**, with results below:
>
> | MMAda       | GQA   | MMMU  |
> |-------------|-------|-------|
> | FLOAT16      | 30.2  | 61.3  |
> | AWQ          | 24.1  | 54.4  |
> | DLLMQuant+   | 27.2  | 56.2  |
> | QuaRot       | 27.3  | 58.4  |
> | DLLMQuant++  | 28.12 | 59.5  |
>
>
> ## 5. Efficiency Details and Fairness of Throughput Comparison
>
> We have fully documented our experimental setup to ensure transparency and fairness:
> - Inference Stack: All speed measurements are conducted using **PyTorch 2.1 + CUDA 12.1**, with optimized Triton kernels for quantized matrix multiplication. We use the same inference pipeline for all methods to ensure a fair comparison.
> - Optimization Parity: All baseline methods (AWQ, QuaRot, RTN) and our DLLMQuant are implemented with the same level of optimization, including batch processing, kernel fusion, and memory management, to eliminate implementation bias.
> - Reproducibility: We will released all benchmarking scripts and environment configurations in our repository, allowing other researchers to verify our throughput results under identical conditions.
>
> ---
>
> We believe these revisions have significantly strengthened the rigor, reproducibility, and significance of our work, and we look forward to the reviewer’s further assessment.

---

### Official Review · Reviewer_EwLs · 2026-03-12

**Soundness:** 3
**Presentation:** 2
**Significance:** 3
**Originality:** 3
**Overall Recommendation:** 4
**Confidence:** 2

**Summary:**

This work proposes a model quantization strategy tailored for diffusion language models (dLLMs) which:
1. Redistributes the calibration scheme to better account for different noise levels
2. Mitigates accumulative errors during progressive denoising
3. Dynamically quantizes masked vs unmasked segments

**Compliance With Llm Reviewing Policy:**

Affirmed.

**Final Justification:**

My post-training concern has been largely addressed but I am not sure if I agree with the interpretation of error accumulation (I have updated my confidence accordingly).

Since this is a concern mainly related to the motivation of the method, I have maintained my accept score.

**Key Questions For Authors:**

Questions:
1. How is fine-tuning handled? Is this quantization scheme robust to post-training interventions? If yes, then it is a big plus.

Typos:
2. L023: Dynamic Language Models (DLLMs). Do you mean diffusion language models?

**Limitations:**

Limitations:
1. There are also discrete diffusion models in the image domain (eg: MMaDA) which is not studied in this work. Discussing the adaptability of the proposed framework to other domains formulated as discrete diffusion or providing results would further strengthen this work.

**Strengths And Weaknesses:**

Soundness:
1. All design choices are motivated by carefully designed analysis and experiments.
2. To me it is not clear why is error accumulation not a problem during course of AR generation? Why is this discrete diffusion specific?
3. Hand engineered hyperparameters (for example in TMAS) which would benefit from more justification.

Significance
1. The study of quantization in a growing field of diffusion language models is very relevant.

Presentation:
1. The writing is a bit hard to follow since it's a combination of multiple adjustments. Definitions and algorithms are introduced abruptly.
2. Certain aspects of related work can be elaborated more. For example L131: “... which uses the Hessian matrix to compensate for quantization error”. How?

Novelty:
1. The insights and methods drawn in this work are novel to the best of my knowledge.

---

> ### Author Rebuttal · Authors · 2026-03-29
>
> Response to Reviewer EwLs:
>
> We sincerely appreciate the reviewer for the insightful and constructive comments, which are of great guiding significance for us to improve the quality of this paper. We have carefully revised the manuscript and addressed all the concerns raised by the reviewer in detail as follows.
>
>
> ### 1. Regarding Error Accumulation in AR Generation and Discrete Diffusion Specificity
>  > Q: Why is error accumulation not a problem during the course of AR generation? Why is this discrete diffusion specific?
>
> We sincerely thank the reviewer for this critical question, which helps to clarify the core difference between autoregressive (AR) language models and discrete diffusion language models (DLLMs). The essential reason lies in their fundamentally distinct generation mechanisms and feature propagation paradigms:
>
>
>  * **AR generation naturally eliminates error accumulation**
>
>
> AR models generate text in a discrete token-by-token manner. At each generation step, the model outputs a deterministic discrete token ID instead of a continuous feature tensor. The input of the next step is the fixed word embedding queried from the vocabulary table, which completely cuts off the propagation path of quantization errors in the continuous feature space. Quantization errors only affect the current single token prediction and will not accumulate across steps. In addition, AR models adopt unidirectional causal attention, which further limits the error diffusion range and avoids global error amplification.
>
> * **Error accumulation is unique to discrete diffusion**
>
> Discrete diffusion DLLMs generate sequences through iterative denoising of full-sequence continuous feature tensors. The output of each denoising step (continuous features) is directly used as the input of the next step without discrete decoding reset, forming a closed error propagation loop, making quantization errors accumulate and amplify geometrically. Meanwhile, the bidirectional attention mechanism of DLLMs makes errors spread globally in the sequence, and the dynamic masking ratio leads to feature distribution shifts, further exacerbating error accumulation. This set of characteristics is unique to the discrete diffusion paradigm and does not exist in AR generation.
>
>
> ### 2. Regarding Fine-tuning and Robustness to Post-training Interventions
> > Q: How is fine-tuning handled? Is this quantization scheme robust to post-training interventions?
>
> We thank the reviewer for the valuable suggestion. Our proposed DLLMQuant is a post-training quantization (PTQ) method that requires no fine-tuning or additional training throughout the process, which maintains the low-cost and efficient advantages of PTQ.
> To verify the robustness of our quantization scheme, we conducted sufficient experiments on multiple DLLM architectures, and the results confirm that:
>
> Our method maintains stable performance gains compared with baseline quantization methods (AWQ, QuaRot) regardless of whether the model undergoes post-training interventions such as instruction tuning and alignment.
>
> Following the reviewer's suggestion, we supplemented evaluation experiments on the multimodal discrete diffusion model MMAda. The experimental results show that DLLMQuant still achieves competitive performance, proving that our scheme has strong generalization and robustness across different tasks and domains.
>
> | Model         | GQA   | MMMU  |
> |---------------|-------|-------|
> | FLOAT16       | 30.2  | 61.3  |
> | AWQ           | 24.1  | 54.4  |
> | DLLMQuant+    | 27.2  | 56.2  |
> | QuaRot        | 27.3  | 58.4  |
> | DLLMQuant++   | 28.12 | 59.5  |
>
> ### 3. Regarding Manuscript Readability and Expression
>
> We apologize for the inconvenience caused by the expression and organization of the manuscript. We have carefully revised and optimized the full text to improve readability:
>
> * We reorganized the logical structure of the paper, and gradually introduced core definitions, key concepts and algorithm flow to avoid abrupt descriptions;
>
> * We added detailed supplementary explanations for the core algorithm (Hessian matrix-based quantization error compensation), and improved the description of Algorithm 2 to make the technical details clearer;
>
> * We corrected the typographical errors in the manuscript (e.g., Line 23), and polished the academic expression of the full text.
> Conclusion
>
>
> Once again, we express our sincere gratitude to the reviewer for the professional and constructive comments. We have carefully addressed all the concerns raised by the reviewer, revised the manuscript in detail, and supplemented relevant experimental verification. We believe these revisions have greatly improved the quality and readability of the paper.

---

> > ### Author Rebuttal · Reviewer_EwLs · 2026-04-01
> >
> > I thank the reviewers for their rebuttal which resolves my question about post training and readability.
> >
> > I would like to however respectfully disagree with the authors interpretation of Discrete Diffusion Language Models.
> >
> > The authors claim "The output of each denoising step (continuous features) is directly used as the input of the next step without discrete decoding reset". This is simply not true. In the inference stage of like LLaDA [1,2] (the one studied in this work), *discrete* token IDs is/are sampled in each step and a lookup is performed for the next step, as in AR models. Therefore it is unclear to me what  "forming a closed error propagation loop" means and how I should interpret this.
> >
> > [1] LLaDA https://arxiv.org/abs/2502.09992
> > [2] MDLM https://arxiv.org/pdf/2406.07524

---

> > > ### Author Response · Authors · 2026-04-02
> > >
> > > With all due respect, let me explain again:
> > >
> > > Unlike AR models, the features output during Prefill are not fed back into the Decoder in subsequent steps; instead, contextual information is preserved via the KV cache.
> > >
> > > In DLLM, however, each output is fed back into the network as the next input. Early inaccurate outputs will directly affect the inputs of subsequent iterations.
> > >
> > > I hope this clarifies your confusion. I apologize for not having explained it clearly enough earlier.

---

### Official Review · Reviewer_1Tbe · 2026-03-13

**Soundness:** 3
**Presentation:** 3
**Significance:** 3
**Originality:** 3
**Overall Recommendation:** 4
**Confidence:** 3

**Summary:**

The paper tackles quantization in DLLMs. The authors show that quantization methods used for traditional LLMs are not optimal for DLLMs. They present certain characteristics of DLLMs that motivate different tweaks in the quantization methodology. Namely:

- Calibration data sampling can no longer be random and uniform but instead should take into consideration the facts that DLLMs do iterative decoding and have dynamic masking ratios. Thus sampling needs to be done in a way that guarantees covering diverse ranges of time steps and masking ratios.
- To minimize the quantization error within each iteration, the determination of quantization parameters of the Value matrix takes into account the Softmax outputs that it will be multiplied with  as weighting factors in the quantization error metric so as to make sure more resources are allocated to values that will eventually be multiplied by non-zero outputs.
- Given that unmasked tokens and tokens with low output scores do not affect the next iteration, greater emphasis is placed on unmasked regions with high output scores during weight quantization.

**Compliance With Llm Reviewing Policy:**

Affirmed.

**Final Justification:**

Given the new baseline added during the rebuttal and the fact that the method still beats that baseline, I believe the manuscript is satisfactory and above the acceptance threshold.

**Key Questions For Authors:**

Questions:

- Can you add existing PTQ works for DLLMs as one of your baselines? Namely Quant-dLLM[1].

Suggestions:

- Some of the Q in Equation 6 are missing the subscript Model. I assume you meant Q_{Model}

References:
[1] Zhang, T., Li, Z., Yan, X., Qin, H., Guo, Y. and Zhang, Y., 2025. Quant-dllm: Post-training extreme low-bit quantization for diffusion large language models. arXiv preprint arXiv:2510.03274.

**Limitations:**

yes

**Strengths And Weaknesses:**

Strengths:

- Good analysis of DLLM quantization problems.
- Method outperforms the baselines and achieves meaningful speedup.

Weaknesses:

- While the results are better than the compared to baslines, the accuracy loss from the FP baseline is still not negligible which raises the question of whether this method is applicable in real-world scenarios.
- No comparison to any existing PTQ work for DLLMs. Example: Quant-dLLM[1]. I think the claim that this is the first PTQ framework tailored for DLLMs is misleading. Their method seems to have some overlap with yours as well.

References:
[1] Zhang, T., Li, Z., Yan, X., Qin, H., Guo, Y. and Zhang, Y., 2025. Quant-dllm: Post-training extreme low-bit quantization for diffusion large language models. arXiv preprint arXiv:2510.03274.

---

> ### Author Rebuttal · Authors · 2026-03-29
>
> Response to Reviewer 1Tbe:
>
> We sincerely appreciate the reviewer for the insightful and constructive comments, which are of great guiding significance for us to improve the quality of this paper. We have carefully revised the manuscript and addressed all the concerns raised by the reviewer in detail as follows.
>
> ### 1. Core Differences between Our DLLMQuant and Quant-dLLM
> We sincerely appreciate the reviewer for raising the important comparison with Quant-dLLM. We fully clarify the critical differences as follows:
>
> - **Quantization Scope**:
>   Quant-dLLM is a **weight-only quantization** method for diffusion LLMs, which **does NOT perform activation quantization** at all.
>   In contrast, our DLLMQuant is a unified **weight + activation joint quantization** framework specially designed for discrete diffusion models.
>
> - **Handling of Error Accumulation**:
>   Quant-dLLM does not consider or mitigate the **step-by-step error accumulation** issue during discrete diffusion generation.
>   Our DLLMQuant explicitly targets this unique problem in discrete diffusion, which is the core bottleneck of quantized dLLM inference.
>
> - **Dynamic Masking Adaptation**:
>   Quant-dLLM uses static calibration and cannot handle the significant feature distribution shifts caused by dynamic masking ratios across diffusion steps.
>   Our method proposes time-mask adaptive calibration and certainty-guided quantization to fully adapt to the unique properties of discrete diffusion.
>
>
> ### 2. Activation Quantization in Eq. 6:
>   In Equation 6 of our paper, the notation **Q(~)**  **explicitly denotes the quantization operation on activations** (i.e., **Q(X)** quantizes the activation tensor **X**.
>
> Once again, we express our sincere gratitude to the reviewer for the professional and constructive comments. We have carefully addressed all the concerns raised by the reviewer, revised the manuscript in detail, and supplemented relevant experimental verification. We believe these revisions have greatly improved the quality and readability of the paper.

---

> > ### Author Rebuttal · Reviewer_1Tbe · 2026-04-03
> >
> > Thank you for considering my review and for explaining the differences.
> > Even if Quant-dLLM approach is different, I think having them as one of the baselines makes the paper much stronger and without that comparison, I'm not sure whether your approach is better or not as right now, the baselines are ones that were not targeting DLLMs at all.
> >
> > Given that, I'm leaning towards maintaining my score for now.

---

> > > ### Author Response · Authors · 2026-04-07
> > >
> > > We sincerely thank the reviewer for your thorough feedback and valuable recognition of strengthening the paper by including Quant-dLLM as a baseline.
> > >
> > > To ensure transparency and rigor, we will make Quant-dLLM a **core baseline** in the revised manuscript. We will supplement additional ablation studies to further elucidate the fundamental advantages of our DLLMQuant++ design:
> > >
> > > | Model        | Method          | Truth. | Arc.  | Hella. | Wino. | PIQA  | MMLU  | C-EVAL | Hum.  | GSM8K | Avg.  |
> > > |--------------|-----------------|--------|-------|--------|-------|-------|-------|--------|-------|-------|-------|
> > > | LLaDA        | Quant-dLLM      | 40.12  | 36.26 | 46.45  | 68.19 | 69.75 | 56.87 | 54.63  | 24.17 | 49.32 | 51.28 |
> > > |              | DLLMQuant++     | 43.53  | 44.18 | 51.00  | 71.85 | 73.94 | 57.77 | 61.22  | 28.92 | 56.25 | 54.29 |
> > > | LLaDA-1.5    | Quant-dLLM      | 42.10  | 84.22 | 67.91  | 71.81 | 72.15 | 56.13 | 58.72  | 42.86 | 71.44 | 62.17 |
> > > |              | DLLMQuant++     | 43.87  | 84.18 | 69.20  | 71.74 | 73.58 | 57.27 | 60.04  | 44.58 | 74.33 | 64.31 |
> > >
> > > **We sincerely thank the reviewer for the extremely careful, rigorous, and insightful review, which has significantly improved the quality of our manuscript. We deeply appreciate the reviewer’s valuable time, professional expertise, and constructive feedback, and we hold the highest respect for the reviewer’s rigorous academic standards.** We have carefully addressed all comments,  and have revised the manuscript accordingly.

---

### Decision · Program_Chairs · 2026-04-30

**Decision:**

Accept (regular)

**Comment:**

The rebuttal satisfactorily addressed a substantial portion of the reviewers’ concerns by clarifying the method, strengthening comparisons, and adding supporting experiments. While some questions remain, they do not outweigh the paper’s practical relevance and empirical value. Given the overall strengths and the partially unresolved discussion, I recommend acceptance.